# ‘I Knew Nothing About Parkinson’s’: Insights into Receiving a Diagnosis of Parkinson’s Disease and the Impact of Self-Management, Self-Care, and Exercise Engagement, from People with Parkinson’s and Family Members’ Perspectives: Qualitative Study

**DOI:** 10.3390/geriatrics10030073

**Published:** 2025-05-25

**Authors:** Leanne Ahern, Catriona Curtin, Suzanne Timmons, Sarah E. Lamb, Ruth McCullagh

**Affiliations:** 1Discipline of Physiotherapy, School of Clinical Therapies, University College Cork, T12 X70A Cork, Ireland; r.mccullagh@ucc.ie; 2Centre for Gerontology and Rehabilitation, School of Medicine, College of Medicine and Health, University College Cork, T12 FN70 Cork, Ireland; catriona.curtin@ucc.ie (C.C.); s.timmons@ucc.ie (S.T.); 3Faculty of Health and Life Sciences, University of Exeter, Exeter EX1 2LU, UK; s.e.lamb@exter.ac.uk

**Keywords:** Parkinson’s disease, diagnosis, self-care, physical activity, physiotherapy, health services, caregiver burden, qualitative study

## Abstract

This paper draws on stories of receiving the diagnosis of Parkinson’s disease, which emerged from a broader narrative study exploring beliefs about exercise and challenges facing people with Parkinson’s disease. Background/Objectives: By interviewing people with Parkinson’s disease (PwPD) and their family members, this paper aimed to gain insights into PwPD’s experiences with diagnosis, its influence on exercise engagement, and access to services in Ireland. Methods: This study employed a qualitative research design, using purposeful and maximum variation sampling. PwPD (varying in age, sex, geographical setting, and disease severity) were recruited from urban physiotherapy services. Semi-structured interviews with 12 PwPD and a group interview with four family members were conducted between November 2022 and January 2023. The interviews were recorded, transcribed, and analysed using thematic analysis. Results: Four themes emerged: (1) firstly, there was disempowerment and emotional shock at diagnosis: PwPD expressed frustration with delays in diagnosis and with how language and empathy affected their ability to cope initially. (2) There was a lack of signposting and services access: a strong need exists for clear information on services and resources to prevent social disengagement. (3) In terms of exercise education and self-management support, PwPD lacked early exercise education and guidance, relying on self-education. (4) With regard to the emotional burden on family caregivers, family members manage care logistics and face emotional burdens, which they try to conceal. Conclusions: The delivery of a Parkinson’s diagnosis could be improved by recognising its psychosocial impact on PwPD and families. Providing clear information on services within weeks of diagnosis was considered crucial. Limited exercise education affected PwPD’s ability to self-manage. Early physiotherapy access is strongly recommended to help delay functional decline and encourage an active lifestyle.

## 1. Introduction

Parkinson’s disease (PD), is a progressive neurodegenerative disorder, affecting millions of people worldwide [1]. For PwPD and their families, the moment of receiving a diagnosis of PD is the beginning of a life-altering journey. It may evoke an array of emotions, from fear and uncertainty to relief or even a sense of validation regarding symptoms long endured. Receiving a PD diagnosis can substantially impact PwPD’s quality of life for many years [2]. This, in turn, possibly influences self-care and engagement in beneficial behaviours, such as exercise. Therefore, understanding the subtleties of this critical moment and its emotional impact is vital not only for healthcare professionals tasked with delivering these diagnoses but also for shaping interventions and support systems that are tailored to the unique needs of people with PD (PwPD).

Advances in medicine have made it possible to manage symptoms and enhance quality of life, but the psychosocial impact of receiving a PD diagnosis remains profound and, at times, overlooked. The time it takes to reach a diagnosis, the referral procedure, how the decision was reached, how it is communicated, the information given and explained, the planned follow-up actions, and the initiation of treatment, are all likely to be factors that impact a person’s experience of diagnosis [3]. Each person’s journey is immensely unique and multifaceted, including adjusting to physical and mental challenges to overcoming the shock of a diagnosis which may impact every aspect of their lives.

Person-centred care has been increasingly important in the treatment of chronic illnesses like PD in recent years [4]. Understanding patients’ needs, preferences, and personal experiences is fundamental to this approach, and serves as the foundation for personalised treatments and support services [4]. By giving voice to the lived experiences of people diagnosed with PD, we can gain invaluable insights into the diverse ways in which the diagnosis shapes their perceptions, priorities, coping mechanisms, and, particularly, their early engagement with self-management, including exercise.

The authors are aware of only four studies that have explored patients’ experiences with receiving a PD diagnosis [5,6,7,8]. Schrag et al. [5], in 2016, surveyed patients’ experiences through the European Parkinson’s Disease Association in eleven European countries (*n* = 1775 patients). Peek [6] (*n* = 37) and Warren et al. [7] (*n* = 6) conducted qualitative interviews with PwPD living in the UK in 2017 and 2016, respectively. Although dated, their findings clearly highlight the shortcomings of the diagnosis experience that need to be addressed and highlight the importance of support systems. More recently, Krieger et al. [8] conducted focus groups and semi-structured interviews with PwPD and family members in Germany (*n* = 52 PwPD, *n* = 29 family members). They identified several issues, including the persistent lack of support for people with Parkinson’s disease in accessing self-management strategies, as well as the inadequate and unstructured provision of information, system orientation, and social awareness [8]. These studies show that, regardless of time or national health provision, people are still finding self-management challenging and need more support.

The initial aim of our qualitative study was to explore PwPD’s relationship with exercise, their motivations, needs, and supports. These first results have already been published in a report [9]. However, the impact of receiving the diagnosis on people’s ability and motivation to manage their Parkinson’s disease emerged as a significantly concerning topic in the initial interviews. This led to an iterative expansion of the topic in later interviews, with these data presented in this current paper.

This paper thus seeks to describe the multifaceted experiences of people receiving a diagnosis of Parkinson’s disease in terms of its impact on the emotional, social, and practical dimensions of their journey but also with respect to their early management of their condition through exercise. Through qualitative analysis of patient narratives and experiences, we aim to uncover common themes, challenges, and resilience factors encountered during and after the diagnosis process. This knowledge may inform a more compassionate and holistic approach that respects the lived experiences of PwPD and their families, ultimately enabling people to face PD with dignity and resilience, and to empower them to self-manage their condition.

Therefore, the aim of this paper is to report the impact of the diagnosis on PwPDs’ ability to self-manage and practise self-care through exercise and to discuss these issues further with family members.

## 2. Materials and Methods

The methodology for this study is described in detail in our previous publication, Ahern et al. [9]. A brief description is as follows.

### 2.1. Design

This study employed a qualitative research design. The study was reported following the consolidated criteria for reporting qualitative research (COREQ) [10] (Appendix A).

To explore PwPD’s relationship with exercise [9], we conducted qualitative interviews sequentially over three stages, with the first stage informing the enquiry of the subsequent stage. Stage One comprised semi-structured one-to-one interviews with PwPD; Stage Two was an interview group with family members/caregivers of a person with PwPD in the study, and Stage Three was an interview group with physiotherapists with experience of working with PwPD. The impact of receiving a diagnosis on a person’s readiness to self-manage their Parkinson’s disease, emerged prominently in the initial interviews with PwPD, and this topic was supported and corroborated during the group interview with family members. The discussions with family members gave a broader understanding of the diagnostic experience, as they are key in supporting the individual through the diagnostic process, and interpreting information received from healthcare professionals. The inclusion of family members provided valuable perspectives on the shared nature of the diagnostic journey, recognising that the impact of a Parkinson’s diagnosis extends beyond the individual to their close relationships and support network. The data from the interviews with physiotherapists did not discuss this topic and, therefore, are not included in this report.

This study was conducted in Ireland, and participants were recruited from the city of Cork and surrounding areas.

### 2.2. Recruitment and Study Sample

Purposeful sampling and maximum difference sampling methods were employed to recruit participants. Inclusion criteria for PwPD were the following: Hoehn and Yahr disease stage 1–3; being able to communicate in English; both males and females; and of varying ages (>50 years) and geographical settings (living in both urban and rural sites). Inclusion criteria for their family members/caregivers were the following: self-reported as a caregiver (without a minimum number of hours of assistance per week), aged > 18 years, and being able to communicate in English. Participants were recruited through physiotherapy primary-care services in urban areas and local support groups, where they expressed an interest in taking part in the study by responding to recruitment posters (either via phone or through their local physiotherapists, who subsequently informed the researcher). To accommodate participants and their ability to attend in person, interviews were offered in person, by phone, and online. Participants were excluded from the study if they could not provide consent, could not communicate in English, or did not have a diagnosis of PD. If eligible, the study was explained to them, and a copy of the participant information leaflet was sent to them by post.

### 2.3. Data Collection

Based on the findings of our systematic review [11], the author’s clinical experience, and feedback from our patient–public involvement (PPI) group, we developed a separate topic guide for the semi-structured 1:1 interviews (PwPD) and for the group interview (families) (Appendix A).

The interviews were conducted by a registered physiotherapist (LA, PhD scholar) who received qualitative research-skills training. The interviews were conducted in a private room in a primary-care centre if held in person and via Microsoft Teams for online interviews. The interviews lasted for a maximum of 60 min (including the group interview). PwPD were offered the option of dyadic interviews for support and prompting if required. The interviewer recorded field notes during the interviews. All interviews were videoed or audio recorded and then transcribed. At the end of the interview, the interviewer completed member checking by reviewing the topics discussed and ensured that the information was interpreted correctly.

The data were analysed on a rolling basis, including self-reflection by the interviewer between interviews. We terminated the recruitment of each stage when data saturation was reached (based on criteria code meaning [12] [no new themes were emerging and the data was beginning to repeat]), and progressed onto the next stage, using the data obtained to inform the next stage’s interview topic guide. The interview process was designed to be interactive and iterative, allowing participants to reflect deeply on their experiences while shaping the direction of the conversation. The use of semi-structured interviews allowed for the emergence of rich, contextually grounded narratives, while reflexivity and participant validation ensured alignment with the lived experiences being explored.

### 2.4. Ethical Considerations

The study received ethical approval from the Clinical Research Ethics Committee of the Cork Teaching Hospitals (ECM 3 (gg) 22 February 2022). The research was conducted between November 2022 and January 2023. All participants provided written informed consent before the interviews. If the interviews were conducted via phone or online, a participant information leaflet and consent form were posted to the participants. Prior to the interview, the participants were given the opportunity to ask further questions, and, if happy to proceed, gave fully informed written consent (return via post prior to the interview) and gave verbal consent on the recorded call. This study was conducted according to the European General Data Protection Regulations (GDPR) No.679 of 27 April 2016. Personal data collected at any time during the study were kept strictly confidential. To ensure confidentiality, the data generated during the study were coded with a participant number. All recordings were stored securely on encrypted, password-protected servers in line with institutional data-governance policies. The participants were informed they had the freedom to address only aspects considered as appropriate. The participants were also informed they were free to withdraw from the study at any point without explanation. Quotes used to illustrate findings were carefully chosen to reflect the participants’ viewpoints, and identifiable information was removed or adapted.

### 2.5. Data Analysis

The data analysis was guided by a constructivist phenomenological methodology, which recognises that individuals construct meaning from their experiences through interaction with their social and cultural environments [13,14]. This perspective aligns with the findings of the study, exploring how PwPD interpret and make sense of their diagnostic journey within their personal, social, and cultural contexts. Constructivist phenomenology emphasises the co-construction of meaning between participant and researcher, acknowledging that the researcher plays an active interpretive role in the analytic process [15] and values participants’ subjective accounts as central to understanding lived experience. This lens was chosen to allow for a rich and situated understanding of the lived experience of diagnosis, where multiple realities and meanings are possible.

To analyse the data, we applied Braun and Clarke’s reflexive thematic analysis framework, which offers a flexible and rigorous method for identifying patterns of meaning within qualitative data. While thematic analysis is not inherently tied to any single theoretical position, it can be used within a constructivist paradigm when the analysis is oriented toward understanding how participants make sense of their experiences [16,17]. The data were manually analysed in Excel (Microsoft Corporation) using Braun and Clarke’s framework [18] for reflexive thematic analysis to allow for the identification of emerging themes. An inductive approach was taken, allowing themes to emerge from the data rather than being imposed by pre-existing theoretical constructs. (1) One member of the research team transcribed each interview, highlighting the key notes in the text, and they reviewed the data with reflexivity. (2) A research team member coded the transcripts line by line and highlighted the text to extract data. The codes were reviewed by a second experienced qualitative researcher and relevant adjustments were made. Finally, an integrated code and corresponding list of quotations were completed. (3) The first researcher developed subthemes and themes from the code list. (4) A second and third member of the research team independently reviewed the extracted themes for overlap and sense-making. (5) The first researcher made any required changes, merging similar topics and splitting large topics where required. This method was repeated until all authors reached a consensus and no new themes appeared. (6) Finally, all three researchers examined the relationship between the themes. This approach was considered appropriate for capturing the nuanced and individual nature of participants’ experiences of receiving a Parkinson’s diagnosis.

The analysis of the results was integrated by verbatim statements which emerged in the transcription of the interviews. Some editing of original verbatim statements was provided to facilitate the comprehension, and pseudonyms were used to replace possible names of persons mentioned. Additionally, words were added in square brackets to improve comprehension, and round brackets were used to indicate nonverbal communication. Text was also highlighted in bold to improve its emphasis.

## 3. Results

### 3.1. Participants

We contacted 32 people (*n* = 20 PwPD, *n* = 12 family members). Sixteen were interviewed, *n* = 12 PwPDs (six females and six males, aged 52–84 years), and *n* = 4 family members/caregivers (three males and one female; none were related) (Table 1). The group interview with the family members was held face to face at their request. One PwPD requested a telephone interview, while two requested a dyad interview (one online and one face-to-face). All dyad interviews were conducted with the spouse. For the dyad interviews, the family members declined to partake in the subsequent group interview with other family members. Of the twelve PwPD, eleven self-reported as being keen exercisers, engaging in regular physical activity (self-reported as achieving the World Health Organisation physical activity guidelines weekly), and one self-reported being an infrequent exerciser (P8). All twelve PwPD were independently mobile, and two used walking aids (P5 and P8). Although all twelve participants were recruited through urban-based primary care services, four lived in rural areas. Baseline characteristics (e.g., mobility, ADLs and main caregiver) were self-reported by the participant. The interviewer had no previous physiotherapist–patient relationship with any of the participants.

The analysis of the data led to four overarching themes demonstrating PwPD’s and family members’ experiences and impact of receiving the diagnosis, education about their condition, knowledge of available services, and the emotional burden on the family members. Each theme is presented from PwPD’s perspectives first, then family member/carers’ perspectives (if applicable). Figure 1 presents the themes and subthemes which emerged from the data. The subthemes and representing quotes are displayed in Table 2.

### 3.2. Theme One: Disempowerment and Emotional Shock at Diagnosis

Receiving a diagnosis of Parkinson’s disease (PD) was described by participants as profoundly disempowering, deeply emotional, and life-altering. Participants vividly recalled moments where empathy was lacking and where the manner of diagnosis delivery had long-lasting effects. Two subthemes were identified: the dehumanising impact of clinical encounters and the critical role of compassionate communication.

#### 3.2.1. Subtheme One: The Dehumanising Impact of Clinical Encounters

People with PD consistently described feeling unheard during early consultations, where healthcare professionals missed opportunities to engage meaningfully. PwPD felt frustrated when a delay in their diagnosis occurred and felt that healthcare professionals were not actively listening to them at the time. P1 powerfully stated, “I went to see three consultants, and I told my general practitioner (GP), and none of them listened… and that was infuriating”. This created a sense of invisibility and loss of agency among PwPD. For some, the clinical delivery felt mechanical, and this resulted in disappointment among PwPD at the lack of empathy portrayed by healthcare professionals. Regarding the experience during the delivery of their diagnosis, they emphasised the importance of supportive language and phrasing, as P4 explained, “[The consultant] is probably so used to it, [the consultant] just [did not] realize that they had just taking my world and pulled the rug from under me. I [could not] speak... Because I had aged. I went in with a [musculoskeletal complaint] and came out with Parkinson’s”.

This impersonal mode of delivery also led to feelings of profound disorientation and psychological shock. As P6 reported, “It was very much like… ‘Well, this is where we are now. And you have Parkinson’s. And you got a good ten years before you really have to worry’”.

This often resulted in the PwPD feeling dehumanised and lacking hope. These moments often marked the beginning of emotional isolation and despair.

PwPD discussed the need for support at the time of diagnosis disclosure. They explained having a family member or friend with them when receiving the diagnosis was important; P4 compared the experience to “quicksand”, and P6 reported that once “you hear whether it’s good news or bad news… you switch off… that’s the advantage of having somebody with you… they’re listening and taking in”.

#### 3.2.2. Subtheme Two: The Critical Role of Compassionate Communication

PwPD expressed a strong desire for emotional acknowledgment during diagnosis disclosure, which was rarely met. P10 poignantly stated, “[The consultant] [did not] have the words to console me, because they are a step above (gestures hand over head to show above) the personal side of it. Like, I just wanted someone to (gestures putting a hand around someone) [put] an arm around me and let me ball [cry]. Because nobody dealt with the shock. It’s a frightening place to be”.

Similarly, family members felt marginalised. One family member (FM2) echoed the experiences of the PwPD with regards to the lack of empathy during the diagnosis. FM2 observed, “When delivering news like that, you need empathy, and you need to give them time, because when it hits them, they are sitting there on their own. I can own imagine what goes through their head (shrugs shoulders)”. Family members like FM1 felt dismissed: “I felt totally dismissed coming out” after questioning the diagnostic process, highlighting gaps in communication and respect.

Others echoed the need for empathy and understanding during diagnosis. P12 reported, “[The consultant] [did not] give me much hope you know. And [the consultant] is finished with me, [the consultant] said “there [is not] much more I can do for you”’.

While all PwPD emphasised the importance of a personalised rapport, P4 reported, “… when you get those words, and the doctor is a highly professional person. But they [the consultant] understands practices, but they [do not] understand me”.

FM1 reported that visits with the consultant are “just business”, explaining the absence of a personalised experience. FM4 further added, “They’re [consultant] writing (demonstrates writing), writing, answering questions, but they are never looking at the person who’s asking those questions”. Family members expressed the importance of including them in the care plan and for healthcare professionals not to assume the family member’s knowledge. FM1 stated, “I think as a carer, you need to be sat down, and you need to have a discussion, you need to be included in the plan, as we are the person looking after them”. Adding to this, FM2 reported, “And I think it’s probably important that they [do not] assume that you know, as the years go on, things change and that you kept up to date [with research] with everything that’s changing. You know, [do not] assume that we already know something”.

These conversations revealed how shocking the diagnosis disclosure was for PwPD and their family members and offered little hope. People felt unsupported as they processed the news. Overall, the diagnosis was perceived not simply as a transfer of medical information but as a deeply emotional and relational event that required more compassionate, person-centred communication strategies. Active listening, self-care, and positive self-management strategies may have fostered hope and empowerment.

### 3.3. Theme Two: Navigating a Void—Lack of Signposting and Service Access

PwPD described a vacuum following diagnosis where clear information, direction to services, and emotional support were largely absent. The participants shared a sense of being left to navigate an unfamiliar path alone, with little help from the healthcare system. This lack of early support not only delayed engagement with important services like physiotherapy or support groups but also heightened anxiety, confusion, and a sense of vulnerability. Two subthemes emerged: the absence of structured information pathways, and the risk of isolation and self-navigation.

#### 3.3.1. Subtheme One: Absence of Structure Information Pathways

People with PD reported receiving little-to-no guidance on available resources, physiotherapy, or support groups. P1 stated simply, “Nobody told me about anything, nobody has ever told me about anything”, while P8 added, “In my own opinion, I think there is a lack of communication”.

This led to a lack of knowledge regarding their condition and the services. PwPD found themselves isolated in managing their condition, with P5 explaining, “I can see why people would be freaked out, you know? People are afraid of it [PD]. Because they [do not] know”. The health system was perceived as reactive rather than proactive, with P6 noting, “If you’re not in the [health] system or your GP is not good, you’re forgotten about”. P5 further added to this by stating, “The biggest hurdle is getting the GP’s and consultants to recommend services. The challenge is how to get referred to anything”.

Some believed the health system should redirect its focus from treatment to prevention. P3 reported, “Throughout the early stages when someone is diagnosed why [are they not] redirected towards exercising, towards classes? The Health Service Executive (HSE) [public health service in Ireland] is going to capture them at some point, but when they do it will be too late, they need to start exercising early”. P7 also highlighted the need for immediate signposting to exercise, “Straight away I would say, go on an exercise program, now! [Do not] wait for 12 months”.

The lack of signposting led to some PwPD independently educating themselves about their condition and the available services. They felt unsupported by the health service as they felt “it [should not] be the individual’s responsibility to go and see what’s out there”.

#### 3.3.2. Subtheme Two: The Risk of Isolation and Self-Navigation

In the absence of structured support, many PwPD were forced to self-educate. P4 reflected on the social consequences: “There needs to be a step in between, everyone should be able to find their village. That’s where it’s lacking; you either go straight to the Parkinson’s society or you’re staying here on your own”. Of the available services, PwPD felt they are not tailored and personalised, with some not suitable for young people. PwPD felt alone, with a lack of peer support. As a result, PwPD are frustrated with the lack of clear care pathways and referrals to health professionals or services.

Family members also echoed the frustrations regarding the lack of signposting. FM2 stated, “You need to be included in the plan… not just assume that we know what’s ahead”. This absence of coordinated care exacerbated feelings of vulnerability and abandonment.

Together, these subthemes highlight a systemic gap in post-diagnostic care that leaves PwPD and their families without the tools, information, or emotional support necessary to engage confidently with disease management. The absence of structured pathways fosters disempowerment, delays proactive care, and compounds feelings of abandonment. Strengthening signposting and personalising support from the point of diagnosis could play a vital role in mitigating early-stage distress and promoting long-term self-management and engagement.

### 3.4. Theme Three: The Missing Link—Exercise Education and Self-Management Support

People with PD consistently identified a critical gap in early care: the lack of timely education on the role of exercise in disease management. While participants recognised the benefits of physical activity later in their disease journey, they described how early misinformation or the complete absence of guidance left them uncertain and hesitant. Many only came to understand the therapeutic value of exercise through self-initiated efforts or informal sources, often years after diagnosis. This gap in professional guidance not only delayed engagement in physical activity but also contributed to unnecessary fear, confusion, and social withdrawal. Two subthemes emerged: early exercise misinformation and missed opportunities, and self-initiated learning as a form of resistance.

#### 3.4.1. Subtheme One: Early Exercise Misinformation and Missed Opportunities

With time, all the PwPD reported awareness of the importance of exercise in the management of PD but, at the time of diagnosis, there was not a focus on its importance with many receiving no information or being encouraged to rest. Several PwPD recalled being discouraged from pursuing physical activity at diagnosis, reflecting outdated medical advice. P3 (diagnosed seven years ago) noted, “At the time I was told to take it easy. [Do not] get wound up and relax for myself and not to get exercise or not to be overly strenuous”. P2 (diagnosed three years ago) similarly reported, “Exercise was never encouraged”.

With that lack of education linked with the risk of social disengagement, P7 reflected, “At the start, I was afraid of moving my body, but I’m not afraid now because I know better”. These early messages contributed to fear around movement, delaying engagement in beneficial activities. P4 stated, “I had worked myself into such a state. I [did not] want to go out”. Only two PwPD reported receiving immediate advice promoting exercise, notably P5, who said, “The consultant was adamant about [exercise] the first day”.

#### 3.4.2. Subtheme Two: Self-Initiated Learning as a Form of Resistance

In response to the lack of guidance, many PwPD pursued their own education. P7 emphasised, “The only [exercise-related] knowledge of Parkinson’s that I got was from a physio and that was about five or six years after I was diagnosed”.

Family members also reported relying on independent research. FM2 said, “I have researched it online; no one sat me down and told me what to expect”. This demonstrates a proactive but burdensome form of self-advocacy to bridge systemic gaps.

The lack of timely early education led to many resorting to self-education to improve their knowledge. All PwPD reported that they would prefer early education, provided within a few months of diagnosis, and, although it was not specified which healthcare professional should provide the information, the PwPD strongly highlighted that the information should be provided rather than them resorting to self-education.

Family members also reported a lack of education regarding the progression and management of the condition, again relying on self-education. They echoed the need for education provision for family members. These conversations highlight the lack of guidance about exercise and physical activity in the early stages of the disease.

This theme highlights a significant disconnect between emerging evidence on the benefits of exercise in Parkinson’s and the early-stage clinical guidance provided to PwPD and families. The absence of timely, structured education on exercise and self-management represents a missed opportunity for early intervention and empowerment. Promoting consistent, early-stage information provision could prevent misinformation, reduce fear, and foster proactive engagement in self-management practices from the outset.

### 3.5. Theme Four: Hidden Burdens—Emotional Impact on Family Caregivers

People with PD acknowledged the emotional burden that is oftentimes placed on their family members, reporting that it is not just the person with PD whose life is affected but also their families. Family members assumed much of the emotional and logistical burden post-diagnosis, often without adequate support. All family members expressed fear for the future. Two subthemes were identified: concealed grief and fear for the future and shifting roles from spouse to caregiver.

#### 3.5.1. Subtheme One: Concealed Grief and Fear for the Future

Family members expressed deep fears about the future and grief for lost aspects of their loved ones’ identity. FM2 noted, “She has always cared for everyone, and in my head, I’m looking down the line and saying the road is going to reverse”. FM3 shared fears regarding long-term caregiving: “I worry what’s down the road, you know? She’s only 72 and I’m 83, who will care for her when I’m gone?”

FM1 articulated the emotional struggle: “I find it incredibly hard…sometimes, I want to scream because I still want to be doing things with him and he stopped talking, little things like that”. FM1 also expressed the need to conceal and hide their emotions from their spouse, “I’m losing the man I’m married to; I’m married 53 years. So, I [do not] have the man that I married. I, like, you could get sad about it, but you [cannot] give up, for them, they need us to the strong”.

#### 3.5.2. Subtheme Two: Shifting Roles from Spouse to Caregiver

Many family members described an identity shift, moving from partner to caregiver. FM1 stated, “I’m very much a carer now, where I [was not] a carer”. FM4 described the increasing dependency: “Her strength is quite good, but her dexterity is awful… like you said, you’re actually caring for the person”.

All family members agreed there is a need for social supports for family members. These conversations highlight the challenge family members face in coping with the changes and associated sense of loss, while trying to support and, at times, care for their family member with PD, to enable them to practise self-care and manage PD. This redefinition of roles added emotional strain and highlighted a critical gap in healthcare support for family caregivers.

These insights highlight the often-invisible toll Parkinson’s disease takes on family caregivers, particularly in the early and mid-stages of disease progression. Emotional burdens, fears for the future, and changing relational roles contribute to a sense of grief and strain that remains largely unacknowledged in current healthcare provision. The findings underscore an urgent need for structured emotional and practical support for caregivers, recognising them not only as informal helpers but as individuals experiencing a parallel journey of adaptation and loss.

## 4. Discussion

This paper aimed to explore the experiences of PwPD in Ireland regarding their diagnosis, its impact on exercise engagement, and their access to relevant services. Through in-depth interviews with PwPD and their family members, the study revealed four key themes: the emotional and disempowering nature of receiving a Parkinson’s diagnosis; a lack of clear signposting to available services; insufficient early education on exercise and self-management; and the emotional and logistical burdens placed on family caregivers. These findings highlight significant gaps in the current care pathway and underscore the need for more empathetic diagnostic communication, timely information provision, and early access to physiotherapy and support services. Acknowledging the emotional impact of receiving a Parkinson’s disease diagnosis is essential, and healthcare professionals are encouraged to adopt a supportive, empathetic, and positive approach during diagnosis disclosure. Providing PwPD with adequate time and support to process this life-changing information is recommended. Following the diagnosis, a prompt follow-up appointment is an important opportunity to share relevant information and signpost available services. Early referral to physiotherapy at the point of diagnosis, rather than waiting for functional deterioration, can empower people with Parkinson’s to engage in self-management strategies, particularly exercise. In addition, healthcare professionals should be mindful of the burden often placed on family caregivers. By building rapport with caregivers, clinicians can offer timely support and refer them to appropriate services to help them in their caregiving role.

### 4.1. Disempowerment and Emotional Shock at Diagnosis

Experience of receiving their diagnosis was mixed. Most PwPD reporting a negative experience, with many being able to directly recall the consultant’s diagnosing words, highlighting the emotional impact of their experience. This experience impacts their ability to self-manage and may delay their motivation to engage in exercise services.

Peek [6] highlighted the importance of creating open dialogue and the need for consultants to modify their consultation style in response to individuals [6]. Even though PD is often not regarded as the “worst” neurological condition [19], receiving a PD diagnosis should not be treated as a routine that ignores the psychosocial implications of this diagnosis. People respond differently to the same news, but regardless of these differences, everyone requires time to process the diagnosis, process their reactions, and express their worries and fears [6]. The participants’ narratives revealed a desire to feel secure at this uncertain time, empowered again during a period of disempowerment, and comforted knowing that a plan was in place for the future when they left the diagnostic appointment.

Krieger et al. [8] conducted a qualitative study in Germany exploring the lived experiences, care provision, self-management, and future support needs of people with Parkinson’s disease (*n* = 52 PwPD, *n* = 29 caregivers/relatives) and their relatives. Through interviews and focus groups, including those newly diagnosed (de novo), they identified major gaps in diagnosis communication, emotional support, and access to structured services. Participants frequently reported feeling abandoned, overwhelmed, and underinformed, with both PwPD and families struggling to navigate care and manage the condition [8]. These findings strongly support the results of our study, highlighting shared experiences of poor diagnostic communication, lack of signposting, and the emotional burden on families. Krieger et al.’s [8] work reinforces the need for empathetic, person-centred care and timely access to self-management resources, particularly exercise education. It validates the call for structured, early-stage interventions to support both de novo PwPD and their families in adjusting post diagnosis. Healthcare professionals should be mindful of the emotional impact that this moment can have on PwPD, acknowledging the need for time to process the information and offering psychological support.

Subramanian et al. [20] highlights the critical importance of delivering a Parkinson’s disease diagnosis with compassion, hope, and clear communication, emphasising that the way a diagnosis is shared can have long-lasting emotional and psychological effects on people [20]. Their findings reinforce the need for clinicians to be mindful of the initial emotional shock, to challenge stereotypical representation of the disease, and to encourage early engagement in self-management strategies [20]. Their findings support the need for clinicians to modify their approach based on the individual’s emotional state and to foster early engagement in self-management strategies. Following a diagnosis, it is crucial to offer the person a space to ask questions and process their emotions, providing tailored information relevant to their condition [20]. This aligns closely with this findings in the current study, where participants described how the delivery of their diagnosis profoundly shaped their early coping mechanism, sense of empowerment, and willingness to engage in exercise and self-care. Embedding these principles into clinical practice could help address the gaps identified in early support and information provision, ultimately promoting better long-term outcomes for PwPD.

Guidelines on breaking bad news (reviewed by Fallowfield et al. [21]) place a strong focus on evaluating the patient’s knowledge and emotional wellbeing, providing information and treatment alternatives, and a contact point soon after their receipt of the diagnosis. Most of these guidelines are centered on terminal illnesses, but more emphasis is being placed on the delivery of bad news in neurology [22,23,24]. How a PD diagnosis is explained to patients still needs to be improved, particularly with regard to remaining sensitive and aware of patient’s understanding, feelings, and expectations. Giving time to ask questions at the time of the diagnosis and providing information that is relevant to each person (e.g., signposting to physiotherapy and/or exercise services) increases patient satisfaction with the diagnostic process [5].

### 4.2. Lack of Signposting and Service Access

All individuals reported receiving little or no information at the time of diagnosis, and many resorted to self-education, but others were at the risk of becoming social disengaged, particularly the younger PwPD (aged < 65 years) as they felt there was no supports that suited their age profile. At the time of diagnosis, they required guided and staged information instead of depending on themselves to learn about their condition. PwPD would welcome early information and signposting to community services for self-management and exercise engagement.

In 2023, O’ Shea et al. conducted a national survey [25] (*n* = 1402) and an independent qualitative interview study [26] (*n* = 22) of PwPD’s experiences of accessing the health service in Ireland, which supports our findings. Experiences included not being advised to bring a family member at the time of diagnosis; an over-emphasis on issues that were non-modifiable, including too much emphasis on loss and degeneration; inadequate signposting and time to discuss available needs/services and self-management strategies; limited written information to take home; and a lack of empathy and privacy (especially for hospital in-patients) [26]. Furthermore, over 75% were not contacted by a healthcare professional in the weeks following diagnosis [26]. An earlier interview study with 19 PwPD and 12 family members in Ireland (2017) [27], which focused on the palliative care needs of PwPD, similarly found an overall lack of information at diagnosis. Participants in that study had suggested that information should be tailored to each person, available in many formats, with additional phone support from a PD nurse for general information or to clarify prior information.

In 2024, Agley et al. [28] explored the physical activity and exercise experiences, preferences, and priorities of PwPD living in the UK through an online survey (*n* = 405). They reported that many participants (68%) had not received education (*n* = 276) or an exercise intervention (54%) (*n* = 217) as part of their routine management by the national public health system since diagnosis; this resulted in seeking services privately [28]. The authors noted that the knowledge of the overall benefits of exercise was good; however, participants lacked specific knowledge on the impact of PD on posture, falls, and muscle strength [28]. Agley’s et al. [28] conclusions echo the findings of this study. PwPD want physical activity and exercise education interventions that provide knowledge, skills, and access to activities [28]. For the most part, these interventions have not been provided as part of their routine care pathway [28]. Physiotherapists reported that education should be provided early as soon as the patient is open to it (during diagnosis may not be appropriate) [9]. To align with the priorities and preferences of PwPD, physical activity and exercise interventions should be offered around the time of diagnosis, including relevant, individualised information and highlighting its value in mitigating PD symptoms [28].

### 4.3. Exercise Education and Self-Management Support

PwPD reported that education about the importance of exercise from physiotherapists was very important. But a delayed referral or access to physiotherapy can result in a delay in receiving this information. It is well recognised that all healthcare professionals promote exercise. However, additional information and support may be needed in the early stages. Surveying healthcare professionals working with PwPD in the UK (*n* = 29 doctors, 17 nurses, and 106 physiotherapists), Agley et al. [29] identified the main educators of physical activity in PD were physiotherapists. This included providing evidence on the role of exercise and signposting to support groups, providing early supports needed to promote exercise self-management.

The model of care from the Irish National Clinic Programme for Neurology [30] has an appendix on PD, which calls for timely and appropriate referrals to services, i.e., it is recommended that PwPD are assessed by relevant, specialised clinicians, with access to the required investigations, and that they have easy access to all the relevant professions, information, and services that enhance the short- and long-term management of their condition [15].

Similarly, the National Institute of Health and Care Excellence (NICE) guidelines on PD advocate that PwPD are seen within six weeks of the referral [31]. However, referrals to physiotherapy services usually only occur when function deteriorates, not around the time of diagnosis [19]. A delayed referral, leading to delayed self-management and exercise guidance, can prevent the opportunity to develop beneficial habitual exercise patterns, optimising both physical and psychological health. This further highlights the need for early referral to physiotherapy services at the point of diagnosis.

### 4.4. Emotion Burden on Family Caregivers

All family members expressed fear for the future and identified the need for supports for family members. Family members often assume the role of caregivers and respond to changes in patient symptoms. These circumstances can impact family members’ mental health, more than their physical health quality [32]. Mc Ardle et al. [33] conducted a pilot study aimed to identify psychosocial factors of PwPD (*n* = 64) and carers (*n* = 40) associated with habitual physical activity. They found that higher carer anxiety and depression were linked to an increase in habitual physical activity volume, whereas poorer self-care among carers was associated with a decrease in activity volume over 18 months [33]. Greater carer strain was associated with longer walking sessions after 18 months [33]. This, therefore, emphasises that if family caregivers do not have a coping mechanism and adequate support from other family members and society, the condition may cause a decline in their health and quality of life [34]. Healthcare professional should discuss with family caregivers the need for personal support and refer them to available services if required. Providing caregivers with the necessary emotional support and information about available services can help reduce their strain and empower them to better support PwPD.

### 4.5. Strengths and Limitations

There is limited research that has investigated PwPD’s experiences with the pathway to diagnosis and its disclosure and how it impacts self-management using qualitative interviews. Current research mainly focuses on the diagnosis experience [6,7] or experiences with self-management practices [8,35] in isolation, leaving a gap in understanding how the experience of diagnosis impacts a person’s ability to self-manage the condition. Moreover, studies that have explored this relationship have often relied on surveys [5], which may lack the depth and richness of insight that qualitative interviews could provide. We believe this method allowed our participants to be open and honest about their experiences and provided us with rich data which may not have been acquired through surveys. The following strategies were employed to ensure the trustworthiness of this qualitative research. The interviewer (LA) is a registered physiotherapist, whose special interest and clinical expertise in PD may have led to informed iterative questioning and credibility. Data analysis was completed by three members of the research team, improving credibility. During data collection, dependability was enhanced and the interviews were video-recorded and transcribed verbatim, with transcripts including non-verbal data in brackets to further illustrate the discussion. A comparison between the transcripts and video recordings was conducted to ensure the data were correctly interpreted. Dependability was enhanced during data analysis; the data were peer-reviewed by members of the research team when collected. An audit trail of the research decisions, changes, and data analysis processes throughout the study were documented. Confirmability was enhanced throughout the data-collection process, with thick audit trails, and three additional research team members continuously reviewed emergent topics through critical peer review.

During data analysis, credibility and dependability were enhanced; independent initial coding was conducted by three members of the research team. After this, a consensus meeting discussing emerging themes took place. This process was repeated after recoding and then a peer-review of the final themes took place.

Several factors hinder the transferability of the findings. One of the main limitations of this research is that the findings of this paper emerged during initial interviews of a study aiming to explore PwPD’s relationship with exercise, their motivations, needs, and supports. Therefore, the topic guide and research questions were not focused on exploring this topic and may have resulted in a mismatch between the research focus and underpinning method of study and the findings in this report. Future studies should employ targeted research questions and tailored topic guides specifically designed to explore the diagnosis experience, service access, and support needs in depth, ensuring alignment between methodology and emergent themes. Secondly, the sample size of PwPD included is small [36]; however, guided by information power versus data saturation [37], we gathered rich data from 12 PwPD with varying genders, years since diagnosis, and living circumstances. We believe that data saturation was achieved with no new emerging themes in the final interviews. Additionally, the sample size for the group of family members was very small and may not capture all experiences. Thirdly, the study mostly represents high-functioning PwPD. Although the recommendations provided by Mutrie et al. [38] were followed and every attempt was made to recruit people with varying exercise adherence, the sample included participants who were more interested/already engaged in exercising, only including study promotion, multiple mailings, and personal engagement [38]. The participants were not initially selected to explore the impact of diagnosis. Therefore, the participant variability must be noted. The participants varied significantly in terms of time since diagnosis (1–17 years); this would have introduced substantial recall bias and variability in the interpretation of the diagnostic experience but it also highlighted the lasting significance of the event. For future studies, we recommend recruiting PwPD diagnosed within the last five years to limit recall bias. Lastly, the participants were recruited from the city of Cork and the surrounding areas, meaning that exploring experiences in a larger sample across multiple clinical services would be important.

## 5. Conclusions

People with PD continue to face similar challenges during their diagnosis experience as those that were reported nearly ten years ago. The findings of this study emphasise the need for healthcare professionals to deliver diagnoses with greater empathy and awareness of the psychosocial impact, ensuring that patients have time to ask questions and receive a follow-up appointment within a few weeks of diagnosis to allow for further questioning and information sharing of available services and supports. Embedding structured follow-up appointments within the early post-diagnosis period could improve patient understanding and connection to essential services. Our findings also highlight the critical role of physiotherapy in supporting exercise engagement and self-management. Ensuring that PwPD are referred to physiotherapy at the point of diagnosis—rather than waiting for functional decline—can empower individuals to adopt active lifestyles and maintain independence. Furthermore, healthcare professionals should provide tailored information to both patients and their families, acknowledging the emotional burden placed on caregivers and referring them to relevant support services to optimise both patient and caregiver well-being. From a policy perspective, there is a need to develop clear care pathways that incorporate early intervention, service signposting, and caregiver support.

## Figures and Tables

**Figure 1 geriatrics-10-00073-f001:**
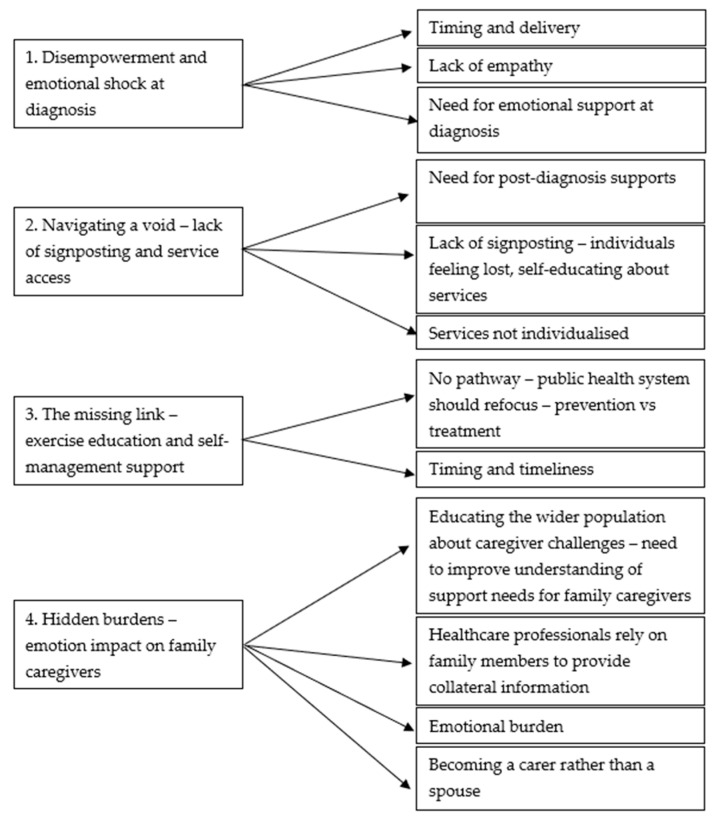
Themes and subthemes.

**Table 1 geriatrics-10-00073-t001:** Clinical characteristics of participants.

**Stage One Participants: People with Parkinson’s Disease**									
**Code**	**Interview Performance Method**	**Age (Years)**	**Sex (M/F)**	**Hoehn and Yahr Stage**	**Years Since Diagnosis**	**Geographical Setting**	**Walking Aid (Yes/No)**	**Activities of Daily Living (Independent/Moderate/Minimum Assistance)**	**Main Caregiver**
P1	In person	57	F	1	4	Rural	No	Independent	Self-care
P2	Phone interview	-	F	3	12	Urban	No	Minimum	Self-care
P3	In person	57	M	1	17	Urban	No	Independent	Self-care
P4	In person	62	F	1	2	Urban	No	Independent	Self-care
P5	Online dyad interview	68	M	2	6	Rural	Yes	Moderate	Spouse
P6	In person	52	F	1	2	Urban	No	Independent	Self-care
P7	In person	84	M	2	8	Urban	No	Independent	Daughter
P8	In person	82	M	2	11	Urban	Yes	Minimum	Spouse
P9	In person	76	F	3	6	Rural	No	Moderate	Spouse
P10	In person	73	F	2	8	Urban	No	Minimum	Self-care
P11	In person	73	M	1	1	Rural	No	Independent	Spouse
P12	In person dyad interview	76	M	3	6	Urban	No	Independent	Spouse
**Stage Two Participants: Group Interview with Family Members (of Participants from Stage One)**									
Code	Sex	Relationship with Stage One Participant		Stage One Participant Code					
FM1	F	Spouse		P8 (11 years diagnosed)					
FM2	M	Spouse		P4 (2 years diagnosed)					
FM3	M	Spouse		P10 (8 years diagnosed)					
FM4	M	Spouse		P9 (6 years diagnosed)					

**Table 2 geriatrics-10-00073-t002:** Themes, subthemes, and quotations.

** *Theme One: Disempowerment and Emotional Shock at Diagnosis* **	
**People with Parkinson’s disease**	
*Subtheme*	*Quotation*
The dehumanising impact of clinical encounters	P1: I went to see three consultants, and I told my GP, and none of them listened… and that was infuriating.
	P4: [The consultant] just [did not] realize that they had just taken my world and pulled the rug from under me.
	P5: They [did not] tell me anything properly, just “you have Parkinson’s”. It felt like being handed a bomb.
The critical role of compassionate communication	P10: I just wanted someone to [put] an arm around me and let me ball [cry]. Because nobody dealt with the shock.
	P6: It was very much like… ‘well this is where we are now’. No offer of help or what I could do.
**Family members**	
*Subtheme*	*Quotation*
Lack of emotional support for PwPD at diagnosis	FM2: When delivering news like that, you need empathy, and you need to give them time.
	FM1: I felt totally dismissed when I questioned the process. Nobody sat us down and explained anything
** *Theme Two: Navigating a Void—Lack of Signposting and Service Access* **	
**People with Parkinson’s disease**	
*Subtheme*	*Quotation*
Absence of structured information pathways	P1: Nobody told me about anything, nobody has ever told me about anything.
	P7: If you [do not] know where to go, you just sit at home, and that makes things worse.
	P8: In my own opinion, I think there is a lack of communication.
Risk of social isolation	P4: There needs to be a step in between, everyone should be able to find their village.
	P9: You have to figure it out yourself. It can feel very lonely.
**Family members**	
*Subtheme*	*Quotation*
Lack of involvement in care planning	FM2: You need to be included in the plan… not just assume that we know what’s ahead.
	FM3: We’re not just drivers and carers. We need to know the plan too.
** *Theme Three: The Missing Link—Exercise Education and Self-Management Support* **	
**People with Parkinson’s disease**	
*Subtheme*	*Quotation*
Early exercise misinformation and missed opportunities	P3: At the time I was told to take it easy. [Do not] get wound up and relax for myself and not to get exercise or not to be overly strenuous.
	P2: Exercise was never encouraged.
	P11: Nobody explained how critical exercise would be; I lost a year just sitting at home.
Self-initiated learning as resistance	P5: I had to look everything up myself—nobody pointed me in the right direction.
	P7: The only [exercise-related] knowledge of Parkinson’s that I got was from a physio and that was about five or six years after I was diagnosed.
**Family members**	
*Subtheme*	*Quotation*
Self-education necessity	FM1: You almost have to become a mini doctor yourself.
	FM2: I have researched it online; no one sat me down and told me what to expect.
** *Theme Four: Hidden Burdens—Emotional Labor of Family Caregivers* **	
**Family members**	
*Subtheme*	*Quotation*
Concealed grief and fear for the future	FM2: I’m looking down the line and saying the road is going to reverse.
	FM3: Who will care for her when I’m gone?
	FM4: You worry all the time, but you try not to show it.
Shifting roles from spouse to caregiver	FM1: I’m very much a carer now, where I [was not] a carer.
	FM4: Her strength is quite good, but her dexterity is awful… like you said, you’re actually caring for the person.
	FM3: We married as equals. Now it feels different; more responsibility, less partnership.

## Data Availability

The datasets presented in this article are not readily available to protect participant confidentiality. Requests to access anonymised partial datasets should be directed to leanne.ahern@ucc.ie.

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
