# Peer review of "‘I Knew Nothing About Parkinson’s’: Insights into Receiving a Diagnosis of Parkinson’s Disease and the Impact of Self-Management, Self-Care, and Exercise Engagement, from People with Parkinson’s and Family Members’ Perspectives: Qualitative Study"

_geriatrics, 2025, doi:10.3390/geriatrics10030073_

Round 1
Reviewer 1 Report
Comments and Suggestions for Authors
Peer review commentary (Manuscript ID: 3584255): I knew nothing about Parkinson’s’: Insights into receiving a diagnosis of Parkinson’s disease and the impact on self-management, self-care and exercise engagement, from patients' and family-members’ perspectives
General comments: This manuscript addresses a critical and timely topic—how people with Parkinson’s disease (PwPD) and their family members experience the diagnosis and early management of the condition. However, the manuscript lacks clarity in both its conceptual focus and methodological rigor. The original study was designed to investigate PwPD’s relationship with exercise, yet this manuscript pivots to explore the impact of receiving a diagnosis. While the theme of diagnosis is indeed valuable, its emergence as a secondary theme in the prior study limits its methodological robustness here. The result is a paper with an ambiguous aim and diluted analytical focus.
Major issues:
- Conceptual confusion and "research drift": The manuscript presents an unclear research objective, oscillating between diagnosis experience and exercise engagement. The two topics, while connected, demand theoretical frameworks. As it stands, the paper lacks a strong central narrative. A clear, singular research question should be developed for future work, prefereable focusing exclusively on "receving a diagnosis".
- Methodology lacks transparency: Essential details are missing from the methodology. Specifically, the process by which participants were informed about the study, inclusion and exclusion criteria, and how and why family members were included are not clearly explained. Furthermore, information regarding the clinical assessments (Hoehn and Yahr stage, Mobility, ADL) is lacking and requires elaboration.
- Participant variability: The participants varied significantly in time since diagnosis (1–17 years), which introduces substantial recall bias and variability in interpretation of the diagnostic experience.
- Underdeveloped results: Thematic analysis seems to be prematurely presented, with insufficient depth. Themes are not thoroughly labelled or substantiated with representative quotations. Greater analytical rigor is required to justify the emergent themes.
- Inclusion of family members: The rationale for including (only) four family members in a single group interview is weak. Their data are presented inconsistently, and there is no discussion of how their insights were triangulated in relation to the PwPD data.
- Literature context and reference gaps: Authors are encouraged to look at studies on de novo Parkinson’s disease patients, which more directly address the psychological and behavioral responses to receiving a diagnosis. Such studies often explore these patients’ initial reflections , providing more immediate and relevant insights on receiving a PD diagnosis. Including this literature would significantly strengthen the paper’s theoretical grounding and interpretative clarity.
Minor issues:
- The abstract should be rewritten to clearly reflect the primary aim and remove conflated objectives (it is my impression that the manuscript combines separate research goals into a single narrative, which weakens the clarity and focus of the study).
Recommendations for authors:
- Consider refocusing the manuscript exclusively on the diagnosis experience, removing the emphasis on exercise engagement (from the aim). This would require clearer methodological justifications and potentially a re-analysis of the data.
- Include more information about how participants were recruited and informed about the study, including ethical and procedural specifics.
- Deepen the thematic analysis with clearly labelled themes and richer quotation use. Select quotations more carefully and consider reducing the number of quotes.
- Incorporate more literature on de novo Parkinson’s disease patients, who can offer timely and relevant perspectives on receiving the diagnosis. These references will better contextualise your findings.
Grammatical issues and typos exist throughout the manuscript (e.g., “emeged” -> “emerged”, “exclusiong” -> “exclusion”). Editing for clarity, grammar, and style is needed.
Author Response
Thank you for taking the time to review my paper. Please see the attachment

Reviewer 2 Report
Comments and Suggestions for Authors
Please, see the attachment.

English language is good, but an overall check for further refinements is suggested
Author Response

(The authors gave the same response as above.)

Round 2
Reviewer 1 Report
Comments and Suggestions for Authors
You have done a good work with the revision. I have no further suggestions for revision.
Author Response
Thank you very much for taking to time to re review this manuscript. We greatly appreciate the time you provided to this.
Reviewer 2 Report
Comments and Suggestions for Authors
Please, see the attachment.

English language is good, but an overall check for further refinements is suggested.
Author Response
Thank you very much for taking the time to re review this manuscript. We greatly appreciate the time you provided to this. Please see below the response to your feedback
Reviewer comment 1: By my opinion, lines 96-110 should be moved to par. 2.1 Design (and however
in the methods), where stages of the study are also mentioned. Also, “Ireland” should be
added in par. 2.1.
Response: Thank you for this feedback. We have movement this information to section 2.1 Design (line 106-121). Additionally we added "Ireland" to par 2.1 (Line 122)
Reviewer comment 2: I would suggest to rewrite lines 126-131 for greater clarity, e.g., as follows:
“Purposeful sampling and maximum difference sampling methods were employed
to recruit participants. Inclusion criteria for PwPD were the following: Hoehn and
Yahr disease stage 1-3; able to communicate in English; both males and females; and
of varying age (> 50 years) and geographical settings (living in both urban and rural
sites). Inclusion criteria for their family members/caregivers were the following: selfreported as a caregiver (without a minimum number of hours of assistance per week),
and aged > 18 years”. Moreover, probably “able to communicate in English” could pertain to caregivers too
Response: We appreciate this feedback, we have amended accordingly (Line 126-132). Also "able to communicate in English" has been added to caregivers (Line 132)
Reviewer comment 3: Data analysis. Line 197: it could be better to write “A research team member”,
instead of “The research team member”.
Response: Thank you, we have changed this in line with your feedback (Line 200)
Reviewer comment 4: 3.1. Participants. Figure 1. The sentences “Enhancing knowledge of greater
population”, and “Family members for collateral” should be explained better
Response: We apologise for not ammending this in the first review. Figure 1 is now amended to provide a more clarity (Line 247)
Reviewer comment 5: Discussion, Clinical Implications. By my opinion, “Clinical Implications” should be merged within the Discussion, and not put as it is at the begin of the section. At least, they could be
sintethysed in the Conclusions.
Response: Thank you for this comment. We have amended as suggested (line 464-475; 504-506; 512-516; 506-509 and 675-680
Reviewer comment 6: To check the whole format of the paper to follow the Journal rules, e.g.: all words in title and headings are capitalized, except conjunctions/articles/prepositions; numbers 1 to 9 are
written in full, except if part of a measurement; to write don't, can't and so on, in full, i.e.
do not, cannot, and so on;
Response: Thank you for this feedback, we have amended as suggested (Line 2-6; 220, 221, 226, 282, 295, 296, 300, 309, 311, 334, 340, 344, 347, 385, 392, 435, 436, 439 and Table 2)
Reviewer comment 7: 4.4. “Emotion burden on family caregivers”, line 577: to put in round brackets only the “n”of the samples, i.e., “….study aimed to identify psychosocial factors of PwPD (n=64) and
carers (n=40)…”.
Response: Thank you, we have amended as per suggestion (Line 598-599)